# Construction of poly-*N*-heterocyclic scaffolds via the controlled reactivity of Cu-allenylidene intermediates

Malla Reddy Gannarapu[1], Takanori Imai[1], Kentaro Iwaki[1], Seiji Tsuzuki[2] & Norio Shibata [1,3✉]

Controlling the sequence of the three consecutive reactive carbon centres of Cu-allenylidene remains a challenge. One of the impressive achievements in this area is the Cu-catalyzed annulation of 4-ethynyl benzoxazinanones, which are transformed into zwitterionic Cu-stabilized allenylidenes that are trapped by interceptors to provide the annulation products. In principle, the reaction proceeds via a preferential γ-attack, while annulation reactions via an α- or β-attack are infrequent. Herein, we describe a method for controlling the annulation mode, by the manipulation of a $CF_3$ or $CH_3$ substituent, to make it proceed via either a γ-attack or an α- or β-attack. The annulation of $CF_3$-substituted substrates with sulfamate-imines furnished densely functionalized *N*-heterocycles with excellent enantioselectivity via a cascade of an internal β-attack and an external α-attack. $CH_3$-variants were transformed into different heterocycles that possess a spiral skeleton, via a cascade of an internal β-attack and a hydride α-migration followed by a Diels—Alder reaction.

[1] Department of Nanopharmaceutical Sciences & Department of Life Science and Applied Chemistry, Nagoya Institute of Technology, Gokiso, Showa-ku, Nagoya 466-8555, Japan. [2] Research Center for Computational Design of Advanced Functional Materials, National Institute of Advanced Industrial Science and Technology, Tsukuba, Ibaraki 305-8568, Japan. [3] Institute of Advanced Fluorine-Containing Materials, Zhejiang Normal University, 688 Yingbin Avenue, 321004 Jinhua, China. ✉email: nozshiba@nitech.ac.jp

Due to their widespread occurrence in nature, indoles and indolines are considered important structural motifs in biologically active molecules, and they are often associated with impressive bioactivity[1–7]. In particular, polycyclic indole/indoline scaffolds that bear multiple stereocenters have received substantial attention from the pharmaceutical industry due to the intriguing drug-like space they present. This is exemplified in particular by the complex molecular structures of alkaloids (Fig. 1a)[8–12]. Conversely, non-natural/artificial organic compounds that possess a trifluoromethyl (CF₃) group at a stereogenic carbon centre, such as efavirenz and DPC 083 (anti-HIV drugs), lotilaner (a veterinary drug), and esaxerenone (a non-steroidal antimineralocorticoid) have been very successful on the pharmaceutical and agricultural markets (Fig. 1b)[13–16]. Efficient methods for the construction of alkaloid-like polycyclic indole scaffolds that contain a CF₃ group at a stereogenic carbon centre would thus be a great advantage for the production of chemically novel drugs[17–22].

To this end, we are interested in cascade annulation reactions using 4-ethynyl benzoxazinanone[23–41]. Xiao, Lu, and co-workers synthesized ethynyl benzoxazinanone in 2016. Using Cu-catalysis, ethynyl benzoxazinanone was converted into the reactive zwitterionic Cu-allenylidene intermediate I via a decarboxylation. I was smoothly intercepted with various sulfur ylides to provide 3-ethynyl-indolines via a [4 + 1] cycloaddition reaction (Fig. 2a)[23]. Ethynyl benzoxazinanone soon became a popular tool for Cu-catalyzed ethynylative annulations. In the presence of suitable interceptors, various ethynyl-N-heterocycles can be constructed from 4-ethynyl benzoxazinanones in a similar fashion (Fig. 2b)[23–41]. That is, the intermolecular annulations start with an attack of the interceptor (Y⁻———Z⁺) at the γ-position of Cu-allenylidene intermediate I followed by the formation of the nitrogen-Z (N-Z) bond and the regeneration of the ethynyl moiety.

Cu-allenylidene intermediates such as I contain three consecutive reactive carbon centres that are labelled, starting from the Cu atom, as the α, β, and γ-positions[42–45]. In principle, the reaction proceeds via mode A and involves a preferential attack of the interceptor at the γ-position of I to furnish the ethynyl-N-heterocycles having two new-bonds (mode A, Fig. 2b)[23–33].

Despite the rich reactivity of Cu-allenylidene intermediates such as I, annulation reactions that proceed via an α- or β-attack rather than a γ-attack have remained scarce[39]. We have hypothesized that annulation mode B involving preferential α- and β-attacks could be realized by controlling the steric and electric factors of suitable substituents X (X ≠ H). Thus, annulation mode B should arise from the pairwise combination of a successive internal β-attack and an external α-attack of I by the interceptor (Y⁻———Z⁺). Subsequent formation of a Cγ-Z bond would provide the non-ethynyl, poly-N-heterocycles with three new-bonds formation (Fig. 2c). During our research into the development of efficient synthetic methods for the synthesis of fluorine-containing heterocyclic compounds for drug discovery[46–57], we noticed that the use of a CF₃ substituent as the X group can direct the reaction pathway from mode A to B[55]. Herein, we realize the idea of a cascade of inter- and intramolecular annulations (mode B), which involves both α- and β-attacks by employing 4-ethynyl-4-CF₃-benzoxazinanones 1 (X = CF₃) and cyclic sulfamate-imines 2 (Fig. 2d; mode B). A wide variety of densely functionalized indoline heterocycles 3 that contain a CF₃ group can be obtained in high yield with excellent diastereoselectivity and enantioselectivity (up to 99% dr and 99% ee). Examples of reactions that generate all-carbon CF₃ quaternary stereocenters at the angular position are extremely rare[58,59], and therefore the obtained results should accelerate corresponding areas of research, especially drug-discovery. The copper-catalyzed asymmetric synthesis via a Cu-allenylidene intermediate attracts much attention[60–62]. Substantial transformations of 3 into more complex molecules are also demonstrated.

The concept of altering the annulation mode by controlling the reactivity of the Cu-stabilized allenylidene intermediate I can also be applied to non-fluorinated substrates. Ethynyl benzoxazinanones 4 that contain a methyl (CH₃) group instead of a CF₃ group are transformed into very different poly-N-heterocycles with a spiral carbazole/indoline skeleton (5) in good yield with high regioselectivity (Fig. 2d; shunt mode B). The unusual formation of 5 occurs via a shunt pathway of mode B that involves the decarboxylative generation of I, followed by a cascade process that involves a cyclization, hydride-α-migration, and a Diels−Alder reaction. The series of spiral poly-N-heterocycles 5

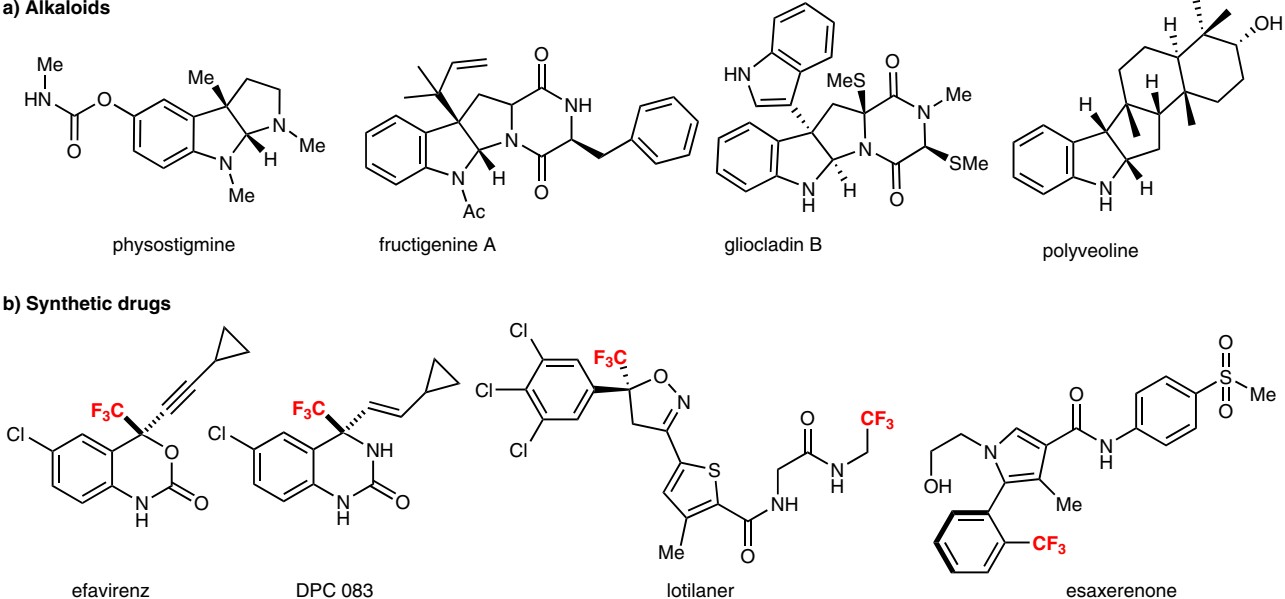

**a) Alkaloids**

physostigmine    fructigenine A    gliocladin B    polyveoline

**b) Synthetic drugs**

efavirenz    DPC 083    lotilaner    esaxerenone

**Fig. 1 Examples of biologically active N-heterocycles. a** Representative biologically active alkaloids with polycyclic indole/indoline scaffolds. **b** Representative synthetic drugs (pharmaceuticals and agrochemicals) with a CF₃ group at a stereogenic carbon centre.

**Fig. 2 Cu-catalyzed decarboxylative annulation of ethynyl benzoxazinanone. a** The seminal work demonstrating the Cu-catalyzed decarboxylative [4 + 1] annulation of ethynyl benzoxazinanone with sulfur ylides. **b** A representative annulation mechanism of ethynyl benzoxazinanones with an interceptor (Y$^-$——Z$^+$) using Cu-catalysis (mode A). **c** Our hypothesis for a different annulation mode (mode B). **d** This work: annulation reactions of ethynyl benzoxazinanones (**1**: X = CF$_3$; **4**: X = CH$_3$) that provide pentacyclic-fused CF$_3$-indolines **3** via mode **B**, and non-fluorinated spiral pentacyclic carbazole/indoline molecules **5** via shunt mode **B**.

generated here are also alkaloid-like indole-rich- molecules. Thus, this method can be expected to serve as a powerful tool for the generation of a drug-like space in a single step.

## Results and discussion

**Annulation reaction using 4-ethynyl CF$_3$-benzoxazinanone 1.**
We commenced our investigation with an annulation reaction of 4-ethynyl CF$_3$-benzoxazinanone **1a** and sulfamate-derived cyclic imine **2a** at room temperature in the presence of CuOTf·1/2 C$_6$H$_6$ (10 mol%), a methyl-substituted Pybox ligand (**L1**, 20 mol%), and i-Pr$_2$NEt (2.4 equiv) in toluene (Table 1, entry 1). To our delight, the reaction proceeded smoothly and delivered the polycyclic indoline (**3aa**) that bears a CF$_3$ group at the all-carbon quaternary centre. However, the yield of **3aa** was only moderate and the enantioselectivity was poor (47% and 15% *ee*). Encouraged by this initial attempt,

we systematically evaluated several chiral ligands (entries 2−4) and found that the phenyl-substituted Pybox ligand **L2** stood out, producing the desired product **3aa** in 70% yield with excellent enantioselectivity 96% *ee* (entry 2). Further results of the other ligands screened, such as **L5** and (*R*)-BINAP are shown in Supplementary Table 1 in the Supporting Information. An investigation into the effect of the solvent on the reaction (Supplementary Table 2) revealed that toluene provides better reaction efficiency than other solvents. Gratifyingly, an evaluation of different Cu salts (entries 5−7 and Supplementary Table 3) revealed that Cu(OTf)$_2$ (5 mol%) and **L2** (10 mol%) resulted in an improved reaction efficiency with a slightly lower yield (68%) and enhanced enantioselectivity (98% *ee*). A slightly improved yield was observed when the reaction was performed with 1.1 equiv of **2a** (entry 8, 71% yield, 98% *ee*; for more details, see Supplementary Table 4). An evaluation of different bases (Supplementary Table 5) showed that i-Pr$_2$NEt was superior to other

**Table 1 Optimization of the reaction conditions for the Cu-catalyzed annulation of 1a with 2a[a].**

| Entry | Ligand | Cu | dr[b] | Yield (%)[b] | ee (%)[c] |
|---|---|---|---|---|---|
| 1 | L1 | CuOTf·1/2C6H6 | >95:5 | 47 | 15 |
| 2 | L2 | CuOTf·1/2C6H6 | >99:1 | 70 | 96 |
| 3 | L3 | CuOTf·1/2C6H6 | - | 28 | - |
| 4 | L4 | CuOTf·1/2C6H6 | >95:5 | 32 | 75 |
| 5 | L2 | Cu(OTf)2 | >99:1 | 71 | 99 |
| 6 | L2 | CuI | - | 21 | - |
| 7[d] | L2 | Cu(OTf)2 | >99:1 | 68 | 98 |
| 8[d, e] | L2 | Cu(OTf)2 | >99:1 | 71 | 98 |
| 9[d, e, f] | L2 | Cu(OTf)2 | >99:1 | 88 | 98 |
| 10[d, e, f, g] | L2 | Cu(OTf)2 | >99:1 | 94 | >99 |
| 11[d, e, h] | L2 | Cu(OTf)2 | - | NR | - |

[a]Reaction conditions: **1a** (0.1 mmol), **2a** (0.1 mmol), [Cu] cat. (10 mol%), ligand (20 mol%), i-Pr2NEt (2.4 equiv) in dry toluene at rt.
[b]Determined by ¹⁹F NMR analysis of the crude reaction mixture using PhCF3 as the internal standard.
[c]Determined by chiral HPLC analysis.
[d]Using Cu(OTf)2 (5 mol%) and **L2** (10 mol%).
[e]Using **2a** (0.11 mmol).
[f]Using i-Pr2NEt (0.5 equiv).
[g]Reaction at 10 °C for 16 h.
[h]Without base.

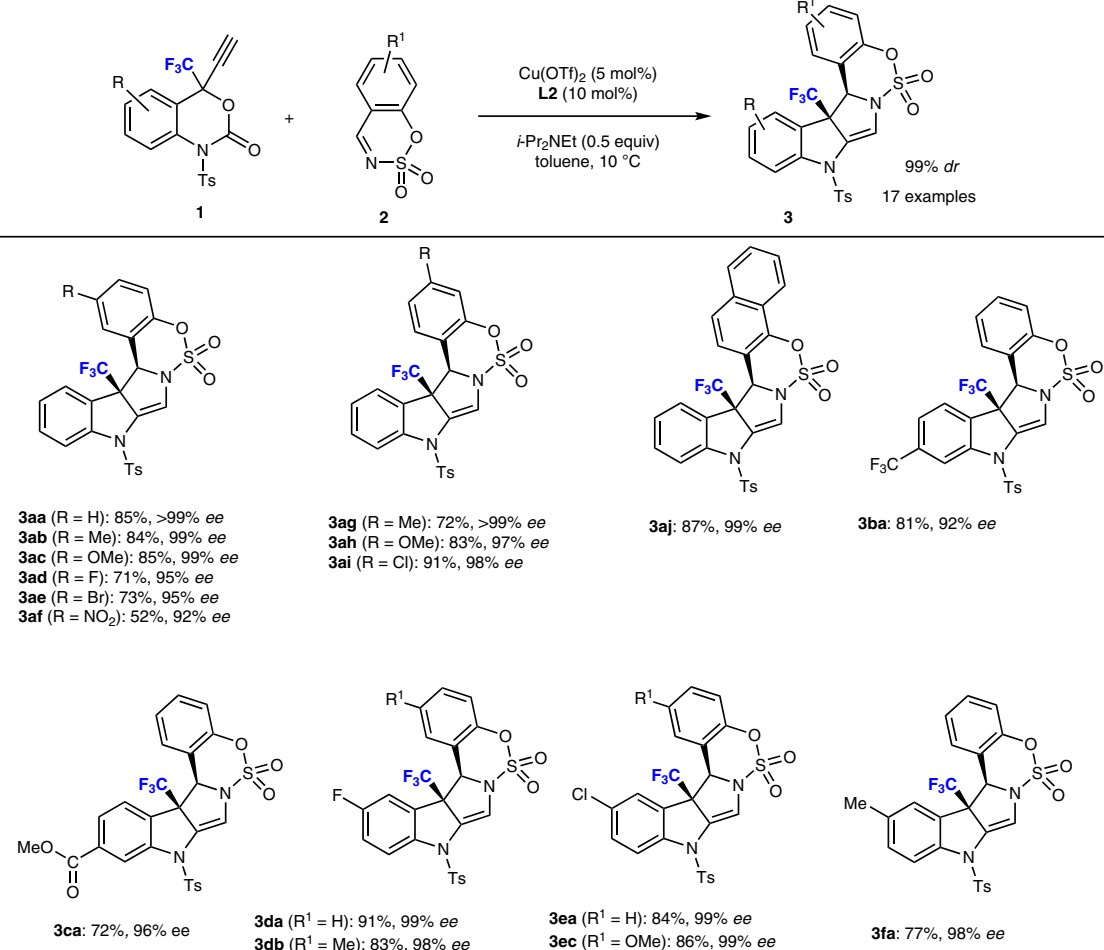

**Fig. 3 Substrate scope for the mode B annulation of 1 with 2.** Reaction conditions: **1** (0.1 mmol), **2** (0.11 mmol), Cu(OTf)$_2$ (5 mol%), **L2** (10 mol%), and *i*-Pr$_2$NEt (0.05 mmol) in 1.0 mL of toluene. The yield values refer to the isolated yield. In all cases, *a* > 99:1 diastereomeric product (**3**) ratio was obtained. The *ee* values were determined by chiral HPLC analysis.

bases and the reaction efficiency was further improved by employing 0.5 equiv of the base (entry 9, 88% yield, 98% *ee*). The most favourable outcome was observed when the reaction was performed at 10 °C (entry 10, 94% yield, >99% *dr*, >99% *ee*). Further experiments revealed that the presence of the base was necessary for this transformation to proceed (entry 11). The absolute configuration of **3aa**, induced by **L2**, was determined by a single-crystal X-ray diffraction analysis (CCDC2026703).

**Substrate scope.** With the optimal catalyst identified and the standard conditions in hand, we studied the scope of the sulfamate-derived cyclic imines **2** for this enantioselective decarboxylative annulation reaction. The results are summarized in Fig. 3. Cyclic sulfamate imines (**2a**-**2i**) that bear a variety of substituents at different positions of the benzene ring, regardless of whether they are electron-donating or electron-withdrawing groups, were tolerated and delivered the annulated products (**3aa**-**3ai**) in good to excellent isolated yield (52−91%) with excellent enantioselectivity (>92% *ee*). The variation of the substituent pattern has thus merely a marginal impact on the selectivity. For instance, substrates that bear halogen substituents such as 6-F (**2d**), 6-Br (**2e**), or 7-Cl (**2i**) reacted smoothly and delivered the desired products in decent yield (71−91%) with respectable enantioselectivity. We observed that the selectivity was slightly decreased from 7-halo (**3ai**; 98% *ee*) substitution to 6-halo (**3ad**, **3ae**; 95% *ee*) substitution. Although 6-NO$_2$ substituted cyclic imine **2f** led to a slightly lower yield, a high enantioselectivity was still achieved in this reaction (**3af**; 52%, 92% *ee*). Moreover, the naphthalene fused cyclic imine **2j** reacted smoothly to produce the desired product in excellent yield and enantioselectivity (**3aj**; 87%, 99% *ee*).

Further experiments were performed in order to evaluate the generality of the reaction. Substituents were introduced at different positions on the benzoxazinanone moiety to create excellent reaction partners and resulted in the desired products with good yield and enantioselectivity. Substrates bearing electron-withdrawing groups, such as 7-CF$_3$ (**1b**), 6-F (**1d**), and 6-Cl (**1e**) reacted efficiently with different sulfamate-derived cyclic imines (**2**) and produced the desired products in good yield (>81%) with excellent enantioselectivity (up to 99% *ee*). Notably, the introduction of an ester group at the 7-position of the benzoxazinanone (**1c**), led to a similar product (**3ca**) in good yield (72%) with excellent diastereoselectivity (>99% *dr*) and enantioselectivity (96% *ee*). Nevertheless, this result should be noted due to the survival of the ester moiety under the applied reaction conditions. The reaction with an electron-donating substituent (CH$_3$) at the 6-position of the benzoxazinanone moiety gave the desired product in a decent yield with optimum enantioselectivity (**3fa**; 77%, 98% *ee*). The stereochemistry of these products was

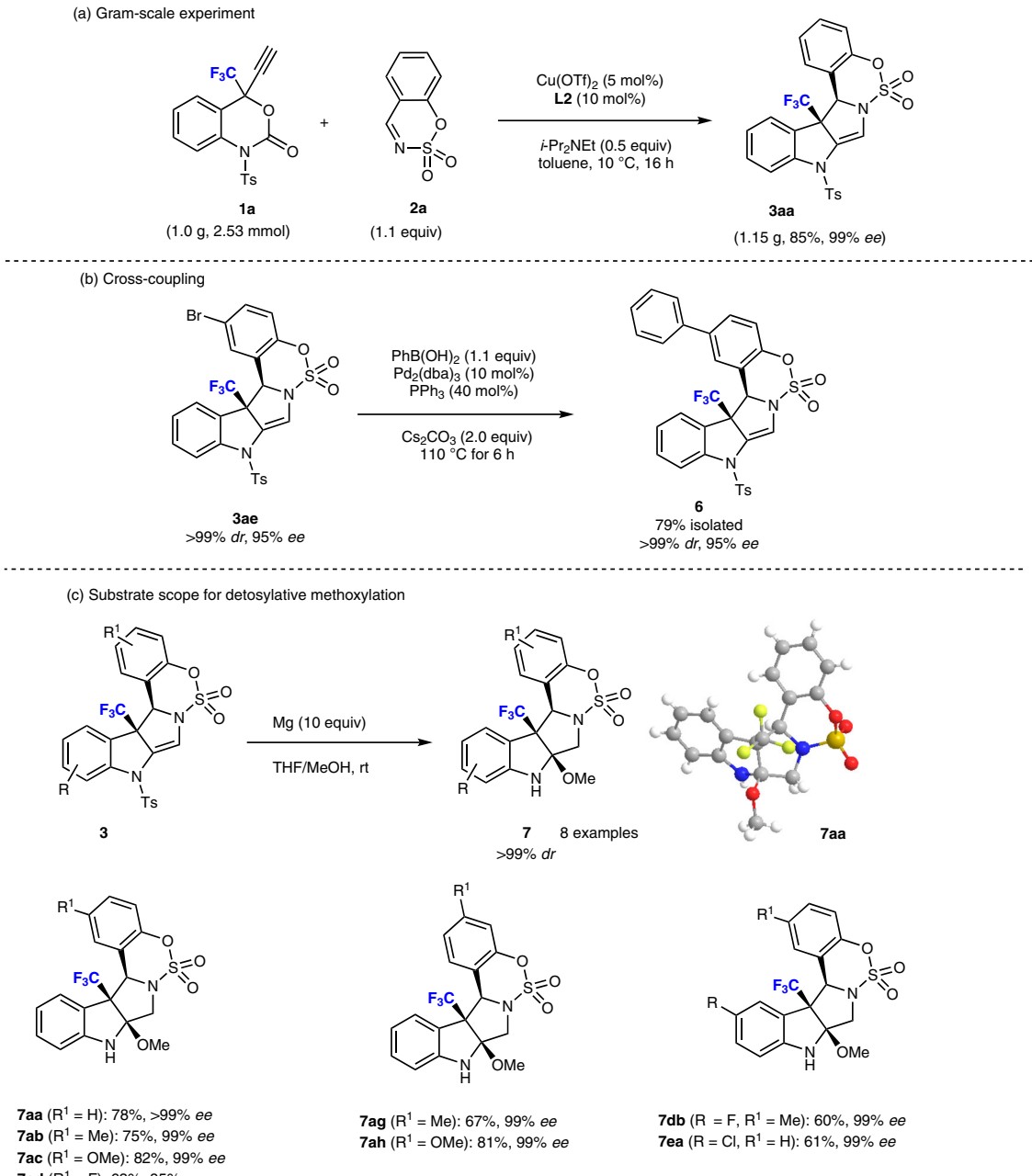

**Fig. 4 Synthetic utility I. a** Gram-scale reaction of **1** with **2**. **b** Cross-coupling reaction using **3**. **c** Detosylative methoxylation of **3** to **7**.

assigned in analogy with **3aa**. In all cases, the diastereoselectivity of **3** was found to be absolute.

**Synthetic utility I.** To further showcase the synthetic potential of this Cu-catalyzed decarboxylative annulation reaction, a gram-scale synthesis of **3aa** was carried out, which achieved an 85% isolated yield without deterioration of the optical purity (Fig. 4a). Gratifyingly, the Pd-catalyzed Suzuki coupling of bromo-substituted-indoline **3ae** with phenylboronic acid afforded biphenyl product **6** in good yield under retention of the enantiopurity (Fig. 4b). As the removal of a *p*-toluenesulfonyl (tosyl) group from an amide usually requires relatively harsh reaction conditions[63], we were concerned prior to attempting the detosylation of **3** due to its high-density functional structure.

Interestingly, treatment of **3aa** with Mg/MeOH under sonication generated another stereocenter, in which successive detosylation/methoxylation reactions occur in a single step and result in the angular methoxylated product **7aa** in 78% yield with outstanding stereoselectivity (>99% *dr*). The absolute configuration of **7aa** was determined by single-crystal X-ray diffraction analysis (CCDC2026704, Fig. 4c). The scope of the detosylative methoxylation was extended to different substrates, and the results are summarized in Fig. 4c. Various substituents on the benzene ring with electronically different properties were well tolerated and gave the corresponding product **7** in moderate to good yield (>63%) with excellent stereoselectivity (up to 99% *ee*). Moreover, halo-substituted indolines (**3db**, **3ea**) afforded the desired products **7db** (60%) and **7ea** (61%) in moderate yield with very good selectivity (99% *ee*). In all cases, the enantiopurity was retained at

**Fig. 5 Synthetic utility II.** Examples of Lewis-acid-mediated functionalization reactions of **7**.

99% *ee* except the case of **7ad**, and the sulfamate moiety remained intact under the detosylation conditions.

**Synthetic utility II**. To further explore the synthetic utility of this transformation, we decided to screen a number of acid-mediated substitution reactions of masked methoxy indole **7aa** with the goal of introducing a substituent at the 2-position (Fig. 5). The Lewis-acid-mediated reaction of **7aa** with allyltrimethylsilane afforded the desired allyl-substituted product **8** in 79% yield with 99% *ee*. Treatment of **7aa** with trimethylsilyl cyanide (Me$_3$Si-CN) and triethylsilane (Et$_3$SiH) delivered the corresponding 2-cyano-product **9** and the reductive product **10** in 88 and 91% yield, respectively, without compromising the enantiopurity. The phosphoric-acid-catalyzed reaction of **7aa** with indole furnished the sterically complex poly-*N*-heterocycle **11** in 82% yield under retention of the enantiopurity. These results suggest that methoxy poly-*N*-heterocycle **7** is a versatile compound that can be easily converted into synthetically challenging di-angular-substituted products in promising yields with optimum enantiopurity.

**Annulation reactions using non-fluorinated, ethynyl CH$_3$-benzoxazinanones**. Next, we attempted the decarboxylative annulation of non-fluorinated, 4-ethynyl-4-CH$_3$-benzox-azinanones **4** with **2** under the reaction conditions optimized for **1** (Fig. 6a). To our great surprise, a mixture of regioisomers of poly-*N*-heterocycles with a spiral carbazole/indoline skeleton **5a** was obtained in 43% yield with high regioselectivity (2C/3C = 85:15). The expected cycloaddition product bearing the **2a** sulfamate moiety was not formed in detectable quantities. The unique, alkaloid-like structure of **5a** led us to investigate the scope of this regioselective transformation of **4** to **5**. We thus treated **4a** under the same catalytic conditions but without the addition of **2a**. As expected, this transformation is generally applicable, and a variety of analogues of **4** were promptly converted into the corresponding spiro-carbazole/indoline molecules (Fig. 6b) in good to high yield (60−79%) with high 2C-regioselectivity (85:15-90:10). The poly-*N*-heterocyclic structure of **5** (2C) was determined unambiguously by single-crystal X-ray diffraction analysis of the brominated spiro-*N*-heterocycle **5c** (2C). The X-ray crystal structure of **5c** (2C) (CCDC2026705) and the HPLC analysis of **5a** conclusively show that **5** is a racemate, which is useful information for the discussion of the reaction mechanism (*vide*

*infra*). The transformation of **4a** also proceeded smoothly with the non-chiral ligand 1,2-bis(diphenylphosphino)ethane (DPPE) to give the same spiro-carbazole/indoline **5a** in similar yield (56%) and regioisomeric ratio (2C:3C = 88:12). The results using other ligands were also attempted and similar results were obtained (Supplementary Table 7). Removal of the tosyl group of **5a** (2C) was achieved with Mg in MeOH/THF to yield spiro carbazole/indole derivative **12** with a 2-CH$_3$-3*H*-indole skeleton (Fig. 6c). Suzuki−Miyaura coupling reaction of **5c** (2C) with phenyl boronic acid (PhB(OH)$_2$) under Pd-catalysis gave the *bis*-coupling product **13** in 56% yield (Fig. 6d).

**Proposed reaction mechanisms**. Based on the experimental results and our own hypotheses, we propose a feasible reaction mechanism to rationalize the formation of polycyclic merged indolines **3** from the reaction of 4-ethynyl 4-CF$_3$-benzox-azinanones **1** with cyclic sulfamate imines **2** (Fig. 7a). Initially, a Cu catalyst (stabilized by its ligands) reacts with **1a** in the presence of a base (*i*-Pr$_2$NEt) to generate Cu acetylide **A**. Subsequently, the decarboxylation reaction of **A** generates zwitterionic Cu-allenylidene intermediate **I**. Due to the presence of the steri-cally demanding CF$_3$ group at the γ-allenyl position, the cyclic sulfamate imine **2a** does not smoothly approach the expected γ-position of **I** for the annulation mode A. A double-helical, pseudo-C2-symmetrical architecture of Cu-complex **I** optimized by DFT calculations is displayed with their selected atomic charge distributions (Fig. 7b, see more details in Supplementary Fig. 1). Ligand **L1** was used for computations. The computed optimized conformation supports the fact that the CF$_3$ group highly blocks the Cγ-position of **I**. Thus, a conventional γ-attack would be unfavourable. Besides, the nitrogen atom is close to the β-carbon of Cu-allenylidene intermediate **I** (N---Cβ: 2.59744 Å). It should be interesting to note that the β-carbon is more positive electronic density than others (Cα: −0.746; Cβ: 0.187; Cγ: −0.149). Annulation mode B should then arise from the pairwise combi-nation of a successive internal β-attack by the anionic amide moiety in zwitterionic Cu-allenylidene intermediate **I** to zwitter-ionic Cu-indoline intermediate **II**, and a [2 + 3] cyclization of **II** and **2a**, respectively. The [2 + 3] cyclization step of **II** and **2a** consists of the formation of a C−C bond between the γ-carbon (Cu-indoline **II**) and the α-carbon (imine **2a**), an external α-attack on **II** by the sulfamate imine **2a** providing the [2 + 3]

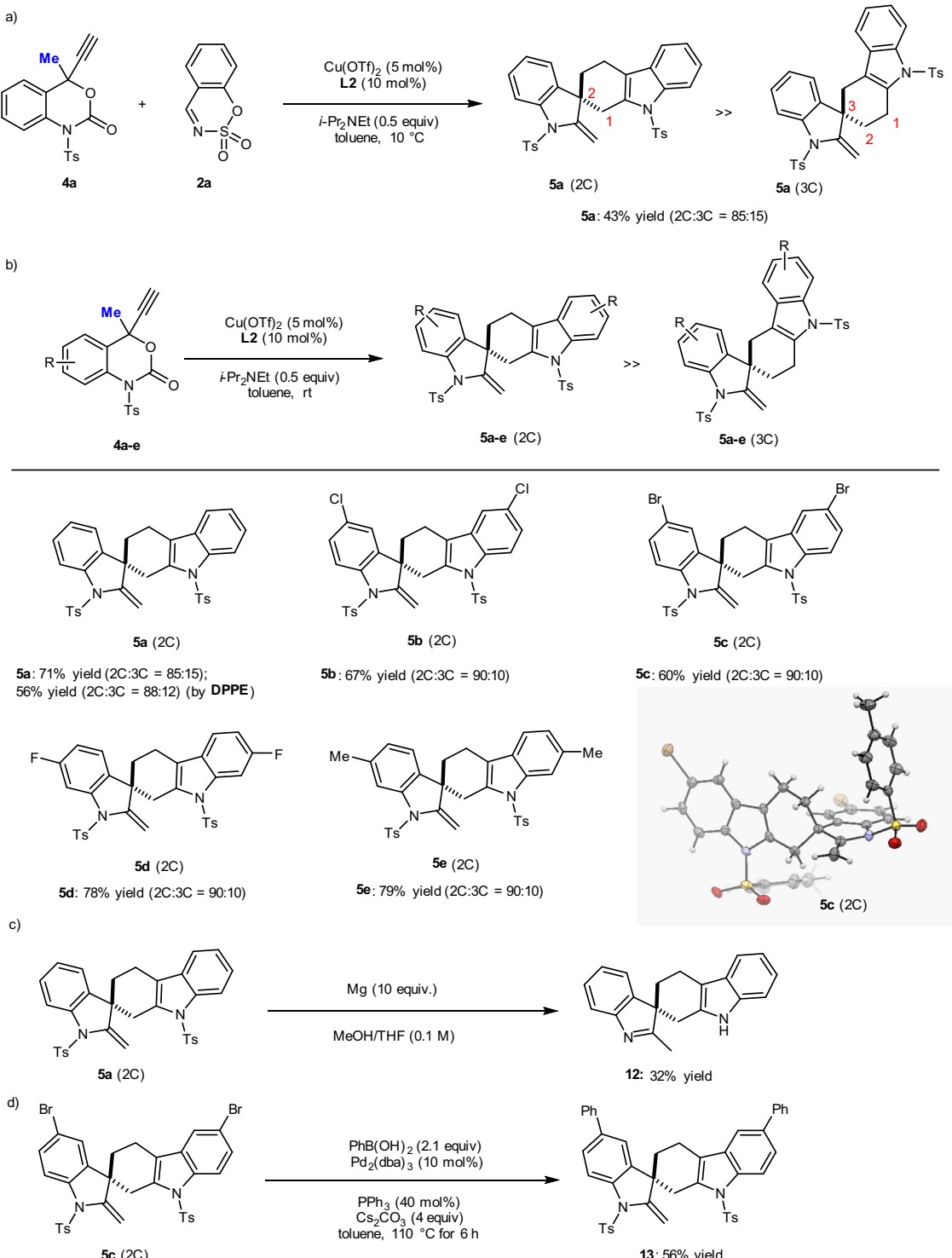

**Fig. 6 Cu-catalyzed decarboxylative annulation of CH₃-benzoxazinanones 4. a** Reaction of **4a** with **2a** under Cu-catalysis. **b** Substrate scope for the Cu-catalyzed decarboxylative formation of spiral carbazole/indoline derivatives **5** from **4**. **c** Detosylation of **5a**. **d** Suzuki−Miyaura coupling reaction of **5c** (2C) with PhB(OH)₂.

annulation product **B** as a Cu-salt. Finally, **3aa** is obtained from protonation of intermediate **B** mediated by *i*-Pr₂NEt, followed by the regeneration of the Cu catalyst in the final step. As illustrated in **I** and **II**, it should be noted that the β-cation of Cu-allenylidene zwitterionic intermediate **I** and γ-anion of Cu-indoline zwitterionic intermediate **II** are likely to be stabilized by negative hyperconjugation induced by a neighbouring CF₃-substituent[64–67]. A newly generated (*S*)-CF₃ at an angular position can be explained by the *Si*-face approach of the sulfamate imine **2a**, based on the optimized conformation of Cu-indoline zwitterionic intermediate **II** (with **L1**) generated by DFT calculations (Fig. 7c).

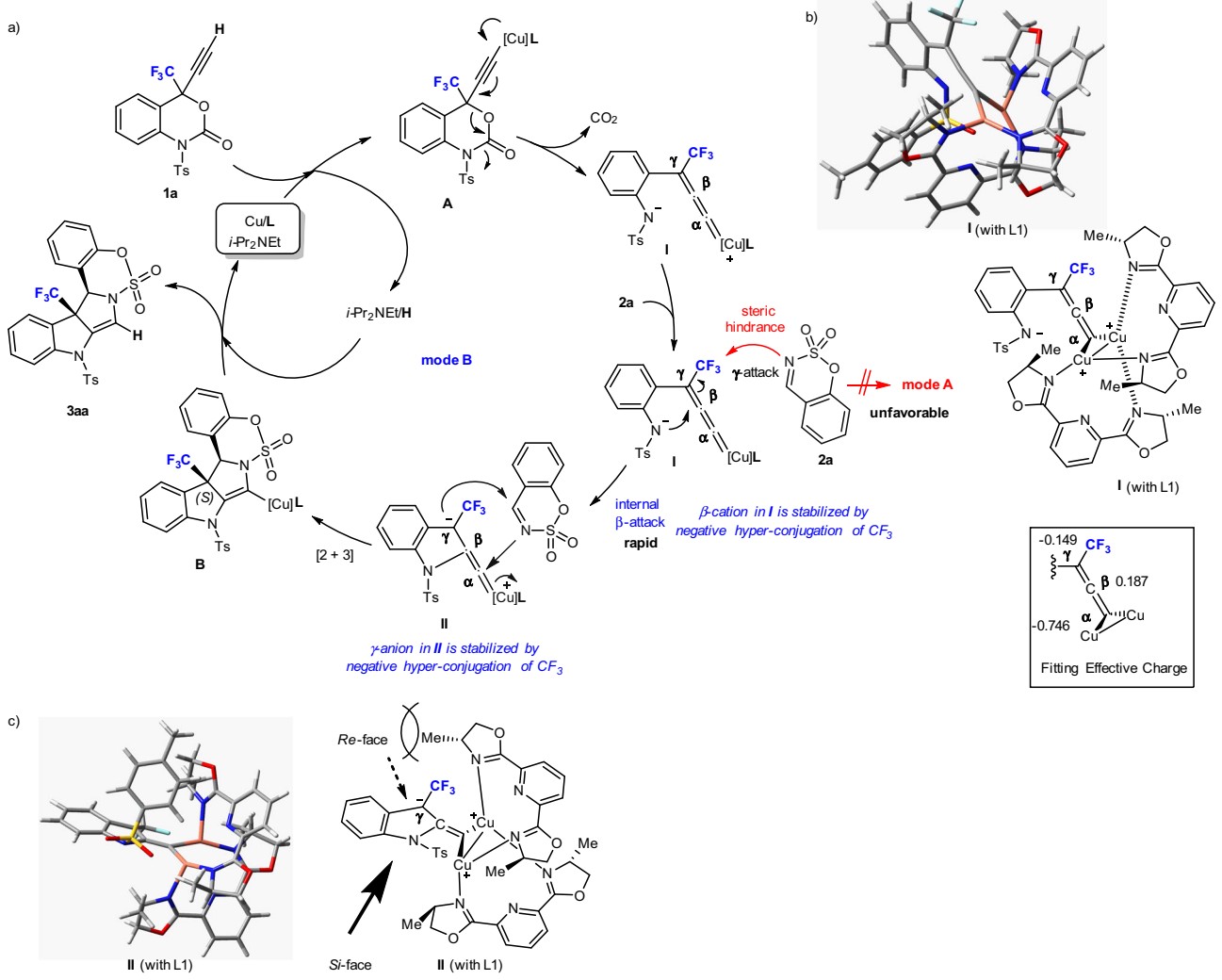

**Fig. 7 A proposed reaction mechanism. a** A Cu-catalyzed catalytic cycle for the decarboxylative annulation of **1a** with **2a** (mode **B**). **b** Optimized geometry of **I** (with **L1**) and calculated atomic charges. **c** Optimized geometry of **II** (with **L1**) and stereochemical model rationalizing the observed stereoselectivity.

The unexpected transformation of the non-fluorinated ethynyl Me-benzoxazinanones 4 into spiro-carbazole/indolines 5 can also be explained based on the proposed mechanism (Fig. 8). The first half of the process for the generation of Cu indoline zwitterionic intermediate II' from 4a via zwitterionic Cu-allenylidene inter-mediate I' is the same as the process for the formation of Cu indoline zwitterionic intermediate II in Fig. 7. Namely, the mode A is unfavourable due to the steric Me group based on the DFT calculation of I' (with **L1**). The nitrogen atom is close to the β-carbon of Cu-allenylidene intermediate I' (N---Cβ: 2.59749 Å), resulting in the intramolecular cyclization to Cu-indoline II. While the γ-anion is stabilized in the $CF_3$-containing inter-mediates I and II by negative hyperconjugation (Fig. 7), that in the non-fluorinated Cu zwitterionic intermediates I' and II' (II'') is unstable due to the positive inductive effect by Me group. The atomic charge distributions are shown in Fig. 8b, c, which indicates that both γ-carbons are relatively positive (Cγ in I': 0.201; Cγ in II' (II''): 0.166). Thus the intermediate II' promptly converts, even in the presence of interceptor 2a, into dimethylene-Cu-indoline C via a cascade process that involves an isomerization to II'' and an intramolecular hydride migration of Me proton to the α-position (shunt-mode B). Cu-dimethylene-indoline C is then protonated to give indole-2,3-quinodimethane intermediate D[68–73] under concomitant release of Cu/L/$i$-Pr$_2$NEt.

D spontaneously dimerizes via a Diels−Alder cyclization pathway to furnish the structurally complex spiro-carbazole-indoline 5a. The lack of asymmetric induction observed in the synthesis of 5a supports a pathway where D spontaneously dimerizes independently from the Cu-catalyzed catalytic cycle (Fig. 8a).

The high regioselectivity of **5a-I** over **5a-II** can be explained by the Alder endo rule[74–76], which involves a favourable interaction between the π systems of the dienophile and the diene (Fig. 8d). While there are two possible endo-modes I and II, endo-mode I is much more suitable in a general HOMO (diene)-LUMO (dienophile)-controlled Diels−Alder process (Fig. 8e)[77–79]. The analyses were also in good agreement with reported concepts of $o$-quinodimethane chemistry[80–83]. Several methods for the generation of indole-2,3-quinodimethane intermediates have been reported;[68–73,84–87] however, they require multistep synth-esis and high reaction temperature. Our method proceeds under mild conditions, and the reaction mechanism is much different from others. This new method for the indole-2,3-quinodimethane generation could be extended for broad applications.

To shed further light on the possibility of controlling the annulation mode by altering the conformation of Cu-allenylidene zwitterionic intermediate **I**, we conducted the reaction of the non-substituted ethynyl benzoxazinanone **14** with two cyclic sulfamate imines (**2a** and **2e**) under the same Cu-catalysis conditions.

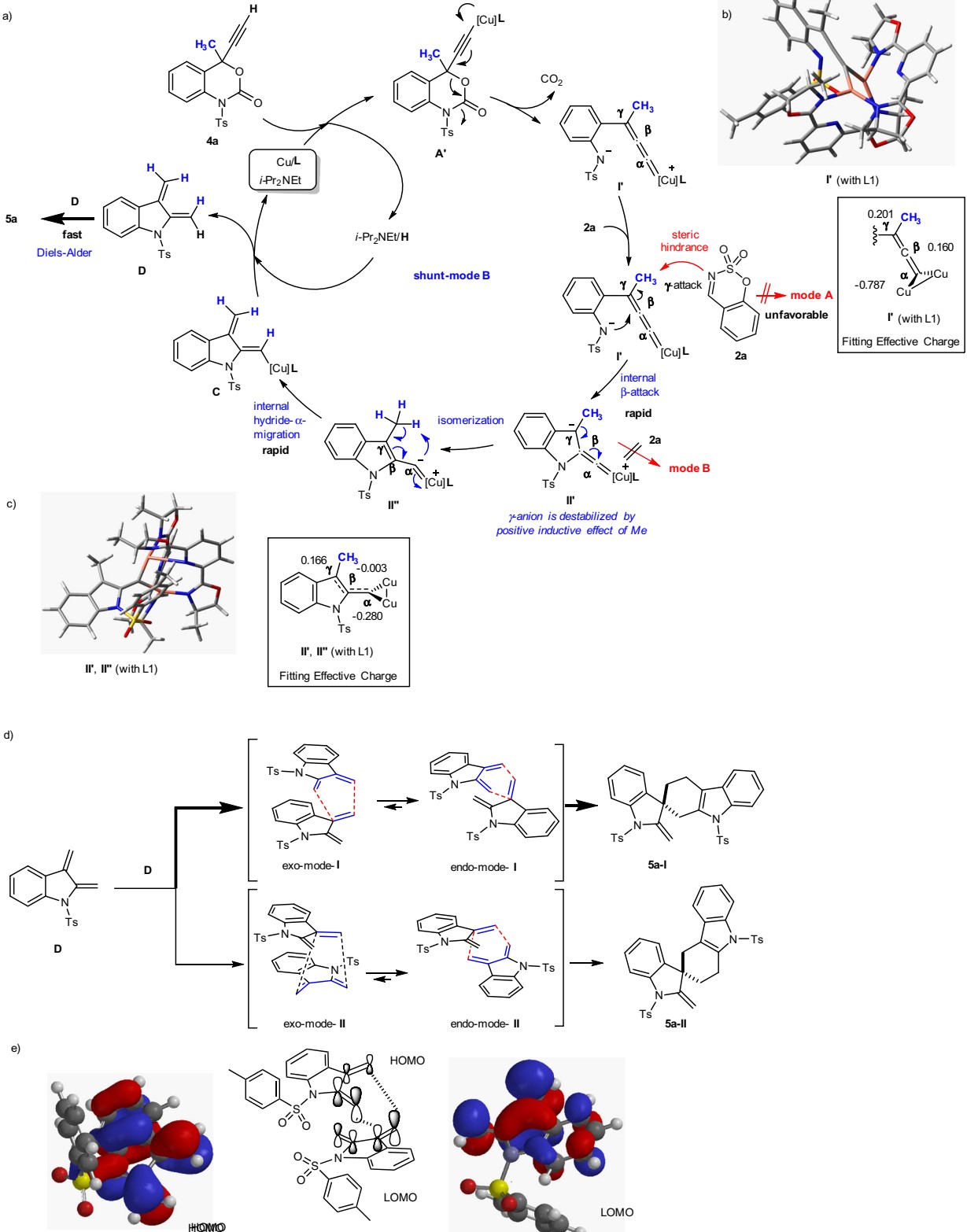

**Fig. 8 Proposed reaction mechanisms. a** A Cu-catalyzed catalytic cycle for the decarboxylative annulation of **4a** (with **2a**) (shunt-mode **B**). **b** Optimized geometry of **I′** (with **L1**) and calculated atomic charges. **c** Optimized geometry of **II′** (**II′′**) (with **L1**) and calculated atomic charges. **d** Regioselective formation of spiro-carbazole-indoline **5a** (2C) rather than **5a** (3C) via an endo-preferential Diels−Alder reaction. **e** HOMO (diene)-LUMO (dienophile)-controlled Diels−Alder dimerization.

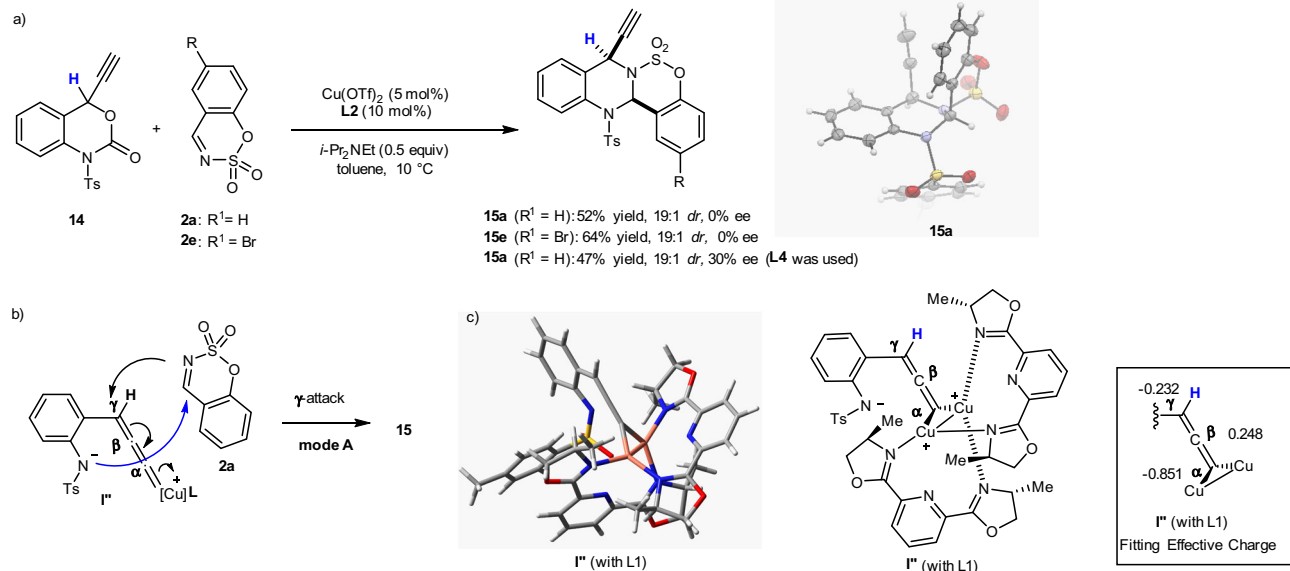

**Fig. 9 Cu-catalyzed decarboxylative annulation of non-substituted 4-ethynyl benzoxazinanone 14. a** Cu-catalyzed reaction of **14** with **2a**, **e**. **b** The reaction mechanism involves mode **A**, not mode **B**, due to the favourable conformation of **I′′′**. **c** Optimized geometry of **I′′**) (with **L1**) and calculated atomic charges.

Remarkably, ethynyl substrate **14** could be transformed, via a Cu-catalyzed decarboxylative [4 + 2] annulation, into two tetracyclic 4-ethynyl-quinazoline derivatives (**15a** and **15e**) in good yield (>52%) with excellent diastereoselectivity (19:1 *dr*) (Fig. 9a). An X-ray diffraction analysis of single crystals of **15a** confirmed a tetracyclic quinazoline skeleton with a *cis*-stereochemical assignment (CCDC2026702). This result is consistent with our proposed mechanism, i.e., that the annulation reaction of the unsubstituted, non-steric, ethynyl **12** with **2** proceeds via mode A, which involves a Cu-allenylidene intermediate **I′′** and a preferential γ-attack of interceptor **2** on **I′′**, to furnish the ethynyl-*N*-heterocycles **15** (Fig. 9b). Mode B, which involves the intramolecular β-attack, was not observed when non-substituted ethynyl benzoxazinanones **14** are used. The optimized conformation of the Cu-allenylidene intermediate **I′′** with the atomic charge distributions (selected) by DFT calculations are shown (Fig. 9c, Supplementary Fig. 2). The γ-position is obviously opened to the nucleophile while the β-carbon is rather sterically shielded, which is in good agreement with the experimental observation mentioned above. Interestingly, however, the β-carbon is the most positive (Cα: −0.851; Cβ: 0.248; Cγ: −0.232). This fact suggests the balance of electronic and steric factor at Cu-allenylidene intermediate is crucial for the reactivity and reaction mode of the annulation reactions.

## Conclusion

In summary, we have demonstrated a strategy for controlling the annulation mode of ethynyl benzoxazinanones based on the reactivity of the Cu-allenylidene zwitterionic intermediate **I** that is formed during the reaction. Via a Cu-catalyzed decarboxylative annulation reaction of 4-ethynyl 4-CF$_3$-benzoxazinanones **1** with cyclic sulfamate imines **2**, densely functionalized indoline scaffolds **3**, which bear a trifluoromethylated all-carbon quaternary centre, were constructed in excellent yield and enantioselectivity (up to 99% *ee*). The key step in the transformation is the unique generation of a Cu-indoline zwitterionic intermediate **II** from a Cu-allenylidene zwitterionic intermediate **I** by intramolecular β-attack. The obtained poly-*N*-heterocycles contain a chiral C-CF$_3$ bond at an angular position; the synthesis of these types of

molecules is usually very challenging. Notably, the chemical transformation of the CF$_3$-poly-*N*-heterocycles **3** resulted in various types of derivatives with excellent selectivity. Significantly, this protocol represents the first example of the construction of optically pure trifluoromethylated merged indoline frameworks. These molecules will most likely become prospective drug candidates. The concept was extended to non-fluorinated, 4-ethynyl 4-CH$_3$-benzoxazinanones **4**. Under the same Cu-catalyzed decarboxylation conditions, CH$_3$-substituted Cu-indoline zwitterionic intermediates **II′** (**II′′**) were generated. The **II′′** was promptly converted regioselectively into the alkaloid-like, spiro-carbazole-indoline derivatives **5** via a cascade process of internal β-attack and hydride α-migration (shunt-mode **B**) followed by a spontaneous Diels−Alder cyclization. This method is generally applicable to a variety of substrates, all of which are attractive for drug-discovery research. Although significant effort has already been devoted to control the reactivity of the α, β, and γ-positions of the Cu-allenylidene, the hitherto reported results are fragmented and no clear strategy has emerged. Our concept, which is based on exploiting the reactivity of the Cu-allenylidene zwitterionic intermediate **I**, may be substantially expanded to synthesize complex polycyclic molecules. The number of permutations and combinations of potential reactants and interceptors that can be used in this cascade annulation will lead to a rich variety of heterocyclic skeletons. Moreover, the concept could be extended to not only use Cu-catalysis but to also use Pd-catalysis and 4-vinyl-benzoxazinanones[87,88]. Further studies on the extension of this concept are currently in progress in our laboratory.

## Methods

**Computational methods**. The ligand framework of zwitterionic Cu-allenylidene intermediates I (X = CF$_3$), I′ (X = Me) and I′′′ (X = H) adopted a double-helical, pseudo-C2-symmetrical architecture according to the reported works of X-ray structure of [Cu$_2$Cl(pybox)$_2$]$^−$[CuCl$_2$]$^{−}$ [89–91], and computed structure of Cu-allenylidene complexed with L1[92]. Ligand L1 was used instead of L2 for computation to simplify their calculations. The geometry optimizations and energy calculations were carried out at the B3LYP/6-311 G** level with Grimme's dispersion correction methods of the D3. Atomic charge distributions were calculated from the B3LYP/6-311 G** level wave functions by electrostatic potential fitting using

the Merz−Singh−Kollman scheme. The optimized geometries and their relative energies are displayed in Supplementary Figs. 1−3. Calculated atomic charges for optimized geometries are shown in Supplementary Fig. 2. The Gaussian 16 programme was used for the DFT calculations in Figs. 7b, c, 8b, c, 9b, c. The references are shown in Supporting Information.

**General procedure for preparation of CF₃-poly-*N*-heterocycles 3**. In an argon filled glove box, a flame-dried 10 mL Schlenk tube was charged with copper(II) trifluoromethanesulfonate (1.81 mg, 0.005 mmol, 5 mol%), 2,6-bis[(4 *R*)-phenyl-2-oxazolin-2-yl]-pyridine **L2** (3.69 mg, 0.01 mmol, 10 mol%) and anhydrous Toluene (1 mL). The resulting solution was stirred for 1 h at 80 °C. In an argon filled glove box, ethynyl benzoxazinanones **1** (0.1 mmol), benzoxathiazine **2** (0.11 mmol), and DIPEA (8.7 μL, 0.05 mmol, 0.5 equiv) were added. The resulting solution was stirred at 10 °C until the complete conversion of ethynyl benzoxazinanones (monitored by TLC). The reaction was quenched by saturated NH₄Cl aqueous solution (2 mL). The resulting solution was extracted with ethyl acetate (5 mL × 3). The combined organic layers were dried over Na₂SO₄, filtered, and concentrated under *vacuum*. The diastereomeric ratio and crude yield were determined by ¹⁹F NMR analysis of the crude reaction mixture. The residue was purified by flash silica gel chromatography (Toluene) to afford the title compound **3**. Full experimental details can be found in the Supplementary Methods.

## Data availability

All characterization data including ¹H, ¹³C, and ¹⁹F NMR spectral data and sample spectra, [α]_D, HRMS, and HPLC chromatograms used to determine enantiomeric purity are included in the Supplementary Information (Supplementary Figs. 4–147). The X-ray crystallographic coordinates for structures reported in this Article have been deposited at the Cambridge Crystallographic Data Centre (CCDC), under deposition numbers CCDC 2026703, CCDC 2026704, CCDC 2026705, and CCDC 2026702. These data can be obtained free of charge from The Cambridge Crystallographic Data Centre via www.ccdc.cam.ac.uk/data_request/cif. The CIF files of CCDC 2026703, CCDC 2026704, CCDC 2026705, and CCDC 2026702 are also included as Supplementary Data 1−4. The electronic structure calculations of **I** (with L1), **I′** (with L1), **II** (with L1), **II′**, **II″** (with L1), and **I″** (with L1), are available as Supplementary Data 5.

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

## Acknowledgements

We thank Hiroto Uno for X-ray crystallographic analyses, and Chika Tanaka for preparing some starting materials. This work was supported by JSPS KAKENHI grants JP 21H01933 (KIBAN B, NS).

## Author contributions

N.S. conceived the concept of this study. M.R.G. optimized the reaction conditions and surveyed the substrate scope. M.R.G., T.I., and K.I. prepared the starting materials. S.T.

examined the DFT calculation. N.S. directed the project. N.S. and M.R.G. prepared the manuscript. The manuscript was written through the contributions of all authors. All authors have given approval to the final version of the manuscript.

## Competing interests

The authors declare no competing interests.
