## [Peer Review File · Communications Chemistry]

Reviewers' comments:

Reviewer #1 (Remarks to the Author):

I very much enjoyed reading this interesting paper and I support its publication in Communications Chemistry. To the best of my knowledge, the synthetic systems studied are all novel. The specific molecular scaffolds are arguably a little esoteric, but nonetheless the observed reactivity, and contrast between CF₃ and CH₃, are very interesting. Significant interest in Cu-allenylidene chemistry more generally also means that the paper will be of interest and value to those working in this area, as well as researchers interested in divergent reaction outcomes. The overall levels of rigour/care taken when preparing the manuscript and adopted throughout the research itself are high. In terms of synthesis, the demonstrated scope, yields, scale ups and ee are all good, and the mechanistic proposals are all reasonable/sensible in my opinion, and consistent with the data reported. I note that I am not sufficiently knowledgeable about the computational chemistry aspects to judge these parts critically, but the best of my (relatively limited) knowledge, these aspects all look ok, and further enhance the paper. I have no scientific recommendations to make at the revision stage and I am happy for the editorial staff to deal with any language/stylistic changes etc, therefore, from my perspective I am happy to recommend acceptance in its current form.

Reviewer #2 (Remarks to the Author):

In this manuscript, Shibata and coworkers has developed an efficient method to construct poly-N-heterocycles through copper-catalyzed cascade reaction between 4-ethynyl benzoxazinones and cyclic sulfamate imines involving a Cu-allenylidene intermediate. In contrast to previous reports that the nucleophiles prefer to attack the γ -position of Cu-allenylidene, the current reaction proceeds through a successive internal β -attack and an external α -attack of Cu-allenylidene due to the stereo effect of the cyclic sulfamate imines. This uncommon pathway renders the approach effective for obtaining poly-N-heterocyclic scaffolds of medicinal interest. The reaction shows significant fluorine effect. When the CF₃-substituted 4-ethynyl benzoxazinones were chosen as the substrates, a cascade annulation pathway was observed to furnish densely functionalized poly-N-heterocycles in high yields with excellent enantioselectivity. While, the corresponding nonfluorinated substrates were used, a spiral skeleton was generated, in which a hydride migration followed by a regioselective Diels-Alder reaction occurred. The difference between these two kinds of substrates is probably ascribed to the negative hyperconjugation induced by the CF₃. To demonstrate the utility of this approach, the authors also conducted several transformations of the resulting trifluoromethylated heterocycles. The methoxylated compounds can serve as a versatile building block for various transformations, providing an efficient way to construct a range of poly-N-heterocycles that are of interest in organic synthesis and medicinal chemistry. DFT calculations provide a reasonable explanation for the regioselectivity and enantioselectivity of the reaction. Overall, this is a nice paper and can be published in COMM-CHEM after minor revision.

1) The fluorine effect dramatically affects the reaction pathways. To demonstrate this fluorine effect, it would be better to replace CF₃ with CF₂H or an electron-withdrawing group, such as CO₂Et or CN to see what type of structure would be formed ?

2) It would be better to cite the following papers for copper-catalyzed asymmetric synthesis via a Cu-allenylidene intermediate: JACS 2015, 137, 2472; Inorg. Chem. 2006, 45, 10043; Chem 2019, 5, 2987.

Reviewer #3 (Remarks to the Author):

The authors report catalytic methodology for the synthesis of poly-N-heterocyclic compounds. The method is based on a copper-catalyzed decarboxylative annulation of 4-ethynyl benzoxazinones with cyclic sulfamate imines (i.e. benzo[e][1,2,3]oxathiazine 2,2-dioxides). Decarboxylative annulations of this type of heterocycles to synthesize several N-heterocycle scaffolds have been previously reported by different groups. However, most of these reactions proceed by an intermolecular nucleophilic attack on the gamma carbon of the copper allenylidene intermediate (called mode A by the authors). Recently, the authors reported that the presence of a trifluoromethyl group at the 4 position of the benzoxazinone core can change the regioselectivity outcome of the annulation with sulfur ylides (ref 55). Based on this previous finding, the authors now report a new mode of annulation (mode B) which involves an internal attack at the beta carbon of the copper allenylidene followed by a stepwise cyclization with the sulfamate imine (electrophilic trapping at the gamma carbon and subsequent attack at the alpha carbon). The reaction proceeds with excellent levels of diastereo- and enantioselectivity and the scope shown is rather broad. The installed functionalities allow for further derivatization of the products. Of note is the detosylation/methoxylation reaction that allows access to a methoxy-2-substituted substrates amenable to further functionalization by acid-mediated transformations. In addition, the authors also report that the replacement of the trifluoromethyl group by a methyl group at the benzoxazinone substrate drives the reaction through a different reaction pathway producing spiro compounds by a cascade involving internal beta attack, isomerization, hydride migration and final dimerization by a Diels-Alder reaction. Plausible mechanisms have been proposed for both transformations and both have been studied by DFT calculations. In conclusion, this paper is of high quality and represents an important contribution. Thus, I recommend publication after addressing the following issues:

- 1) In Figure 2a, the sulfur ylide should be shown after the formation of the copper allenylidene for better clarity.
- 2) In table 1, it is not clear the amount of copper and ligand used in entries 5 and 6. If 10 mol% was used, something like [Cu] should be written instead of CuOTf·1/2C6H6 in footnote a.
- 3) The enantiopurity of product 3ca was not determined (I guess that chiral separation of enantiomers could not be found). Derivatization of this product (e.g. reduction of the ester moiety) should be carried out in order to find a proper derivative that allows determination of the enantiomeric excess.
- 4) Since formation of spiro-compounds 5 proceeds in a racemic manner and can be promoted by the use of non-chiral ligands such as dppe, it would be interesting to add a small screening of this type of ligands for this transformation.
- 5) Experimental section in supporting information is well prepared, and all products are thoroughly characterized. However, integral values of NMR spectra should be checked since some of them are incorrect (see e.g. compound 6 in page S121).
- 6) In their explanation of the reaction mechanisms, the authors say that some carbons are more "cationic" than others in the copper allenylidene intermediate. I would suggest to rather talk about carbons that have more positive electronic density than others.

Reviewer #1: I have no scientific recommendations to make at the revision stage and I am happy for the editorial staff to deal with any language/stylistic changes etc, therefore, from my perspective I am happy to recommend acceptance in its current form.

Response: Thank you for your high evaluation for acceptance.

Reviewer #2:

1) The fluorine effect dramatically affects the reaction pathways. To demonstrate this fluorine effect, it would be better to replace CF₃ with CF₂H or an electron-withdrawing group, such as CO₂Et or CN to see what type of structure would be formed ?

Response: Thank you for your comments. I agree with your suggestions that I am also curious about using other groups such as CF₂H, CO₂Et, or CN to see what type of structure would be formed. The preparations of these starting materials are challenging, and I will report these issues as the next project in the near future.

2) It would be better to cite the following papers for copper-catalyzed asymmetric synthesis via a Cu-allenylidene intermediate: JACS 2015, 137, 2472; Inorg. Chem. 2006, 45, 10043; Chem 2019, 5, 2987.

Response: Thank you for your suggestions. The suggested references are cited (Refs 60-62) with sentences in the main text.

Reviewer #3:

1) In Figure 2a, the sulfur ylide should be shown after the formation of the copper allenylidene for better clarity.

Response: Thank you for your comments. Figure 2a was revised as you suggested.

2) In table 1, it is not clear the amount of copper and ligand used in entries 5 and 6. If 10 mol% was used, something like [Cu] should be written instead of CuOTf·1/2C₆H₆ in footnote a.

Response: Thank you for your comments. It was revised and mentioned as [Cu] cat., instead of CuOTf·1/2C₆H₆ in footnote a.

3) The enantiopurity of product 3ca was not determined (I guess that chiral separation of enantiomers could not be found). Derivatization of this product (e.g. reduction of the ester moiety) should be carried out in order to find a proper derivative that allows determination of the enantiomeric excess.

Response: Thank you for your comments. After additional efforts to determine the enantiopurity of product **3ca**, the enantiomeric excess of **3ca** was determined to be 96% ee by chiral HPLC using CHIRALPAK® IC (n-hexane/isopropanol = 95.0/5.0, flow

rate 1.0 mL/min, $\lambda = 254$ nm) t (major) = 32.133 min, t (minor) = 39.2 min). It was added in the revised manuscript and supporting information.

4) Since formation of spiro-compounds 5 proceeds in a racemic manner and can be promoted by the use of non-chiral ligands such as dppe, it would be interesting to add a small screening of this type of ligands for this transformation.

Response: Thank you for your valuable comments. We now added the results of ligand screening in the main text and Scheme S1 in supporting information.

5) Experimental section in supporting information is well prepared, and all products are thoroughly characterized. However, integral values of NMR spectra should be checked since some of them are incorrect (see e.g. compound 6 in page S121).

Response: Thank you for your valuable comments. The supporting information was re-checked and revised thoroughly.

6) In their explanation of the reaction mechanisms, the authors say than some carbons are more "cationic" than other in the copper allenylidene intermediate. I would suggest to rather talk about carbons that have more positive electronic density than others.

Response: Thank you for your valuable comments to select the right words. The statements were revised as suggested.

REVIEWERS' COMMENTS:

Reviewer #2 (Remarks to the Author):

The authors have fulfilled the requests raised by the reviewers. The current version is suitable for publication in communications chemistry.

Reviewer #3 (Remarks to the Author):

The authors have successfully addressed all the issues raised by this reviewer. Thus, publication is recommended.

REVIEWERS' COMMENTS:

Reviewer #2 (Remarks to the Author):

The authors have fulfilled the requests raised by the reviewers. The current version is suitable for publication in communications chemistry.

Answer:

Thank you.

Reviewer #3 (Remarks to the Author):

The authors have successfully addressed all the issues raised by this reviewer. Thus, publication is recommended.

Answer:

Thank you.